# The Effects of Irrigation Water Salinity on the Synthesis of Photosynthetic Pigments, Gas Exchange, and Photochemical Efficiency of Sour Passion Fruit Genotypes

**DOI:** 10.3390/plants12223894

**Published:** 2023-11-18

**Authors:** Francisco Jean da Silva Paiva, Geovani Soares de Lima, Vera Lúcia Antunes de Lima, Weslley Bruno Belo de Souza, Lauriane Almeida dos Anjos Soares, Rafaela Aparecida Frazão Torres, Hans Raj Gheyi, Luderlândio de Andrade Silva, Francisco Vanies da Silva Sá, Valeska Karolini Nunes Oliveira de Sá, Smyth Trotsk de Araújo Silva, Reynaldo Teodoro de Fátima, Pedro Dantas Fernandes, Alan Keis Chaves de Almeida

**Affiliations:** 1Academic Unit of Agricultural Engineering, Federal University of Campina Grande, Campina Grande 58430-380, PB, Brazil; francisco.jean@estudante.ufcg.edu.br (F.J.d.S.P.); vera.lucia@professor.ufcg.edu.br (V.L.A.d.L.); weslley.bruno@estudante.ufcg.edu.br (W.B.B.d.S.); rafaela.aparecida@estudante.ufcg.edu.br (R.A.F.T.); hans.gheyi@ufcg.edu.br (H.R.G.); luderlandio.andrade@estudante.ufcg.edu.br (L.d.A.S.); valeska.karolini@estudante.ufcg.edu.br (V.K.N.O.d.S.); smyth.trotsk@estudante.ufcg.edu.br (S.T.d.A.S.); reynaldo.teodoro@estudante.ufcg.edu.br (R.T.d.F.); pedro.dantas@professor.ufcg.edu.br (P.D.F.); 2Academic Unit of Agrarian Sciences, Federal University of Campina Grande, Pombal 58840-000, PB, Brazil; lauriane.almeida@professor.ufcg.edu.br; 3Agricultural Sciences Center, State University of Paraíba, Catolé do Rocha 58884-000, PB, Brazil

**Keywords:** abiotic stress, genetic variability, *Passiflora edulis Sims*, Brazilian semi-arid region

## Abstract

The objective of this study was to evaluate the synthesis of photosynthetic pigments, gas exchange, and photochemical efficiency of sour passion fruit genotypes irrigated with saline water under the conditions of the semi-arid region of Paraíba state, Brazil. The experiment was conducted at the experimental farm in São Domingos, PB. A randomized block design was adopted, in a 5 × 3 factorial scheme, with five levels of electrical conductivity of irrigation water—ECw (0.3, 1.1, 1.9, 2.7, and 3.5 dS m^−1^)—and three genotypes of sour passion fruit (Gigante Amarelo—‘BRS GA1’; Sol do Cerrado—‘BRS SC1’; and Catarina—‘SCS 437’. The increase in the electrical conductivity of irrigation water negatively affected most of the physiological characteristics of the sour passion fruit at 154 days after transplanting. Significant differences were observed between sour passion fruit genotypes when its tolerance was subjected to the salinity of irrigation water. There was an increase in the percentage of damage to the cell membrane with the increase in the electrical conductivity of irrigation water, with maximum values of 70.63, 60.86, and 80.35% for the genotypes ‘BRS GA1’, ‘BRS SC1’, and SCS 437’, respectively, when irrigated with water of 3.5 dS m^−1^. The genotype ‘BRS Sol do Cerrado’ showed an increase in the synthesis of photosynthetic pigments when irrigated with water of 3.5 dS m^−1^, with maximum values estimated at 1439.23 μg mL^−1^ (Chl *a*); 290.96 μg mL^−1^ (Chl *b*); 1730.19 μg mL−1 (Chl *t*); and 365.84 μg mL^−1^ (carotenoids). An increase in photosynthetic efficiency parameters (F_0_, Fm, and Fv) of the genotype ‘BRS Gigante Amarelo’ was observed when cultivated with water with high electrical conductivity (3.5 dS m^−1^).

## 1. Introduction

Sour passion fruit (*Passiflora edulis* Sims) is a crop of great social and economic importance and can be used both in the form of food and for medicinal purposes, with fruits rich in several nutrients, vitamins, amino acids, and dietary fiber [1]. The global demand for sour passion fruit juice is growing, with an estimated increase of 15% to 20% per year [2].

In 2021, Brazil produced 683,993 tons of fruits, with the northeastern state standing out among the main producing regions, accounting for 69.6% of the national production [3]. Among the northeastern states, Bahia (207,448 t) and Ceará (177,291 t) stand out as the largest producers, while the state of Paraíba produced only 10,444 t in that year. Although the northeastern region is the largest producer, the average yield per hectare is only 14,758 kg ha^−1^, which is below the Brazilian average (15,259 kg ha^−1^).

Factors that may be related to the low yield in the northeastern state include the edaphoclimatic conditions of this region. High evapotranspiration rates and low rainfall, distributed irregularly in the atmosphere, are intrinsic conditions that force producers to use groundwater, which contains high levels of dissolved salts [4].

Salt stress creates conditions of imbalance in plants by limiting physiological and biochemical processes [5,6]. Under salt stress conditions, the first effect to occur is osmotic, characterized by the restriction in water availability to plants, due to the reduction in the osmotic potential of the soil solution [7]. Then, the excessive accumulation of ions in plant tissues leads to ionic stress, which induces morphological, physiological, biochemical, and molecular changes in cell metabolism [8].

In general, fruit crops have greater sensitivity to salt stress than perennial crops, and their success in agricultural production depends on the use of alternative practices, including the use of salt-tolerant species [9,10]. Salinity tolerance mechanism in plants depends on other factors, such as the capacity for transport and accumulation of toxic ions, activation of the antioxidant defense system, and biosynthesis of compatible solutes [11,12].

Studies with various crops, such as sesame [13], cotton [14], cowpea [15], and sunflower [11], have been carried out to evaluate the tolerance of different genotypes to salt stress, with genotypic variations being observed regarding the tolerance of the crops to the salinity of irrigation water. However, research with fruit species is scarce in the literature, especially with the sour passion fruit crop under the semi-arid conditions of the northeastern Brazilian state. In this context, the hypothesis which was tested was that the effects of saline stress differ in sour passion fruit genotypes, with various impacts on the synthesis of photosynthetic pigments, gas exchange, and photochemical efficiency of the plants resulting from their genetic variability. Therefore, the identification of genotypes tolerant to salt stress is important for the adoption of appropriate agronomic strategies for agricultural production in conditions where only waters with higher salt contents are available. In view of the above, the objective of this study was to evaluate the synthesis of photosynthetic pigments, gas exchange, and photochemical efficiency of sour passion fruit genotypes irrigated with saline water under the conditions of the semi-arid region of Paraíba, Brazil.

## 2. Results

### 2.1. Analysis of Principal Components and Multivariate Variance of the Effects of Salt Stress on the Physiology of Sour Passion Fruit Genotypes

The multidimensional space of the original variables was reduced to two principal components (PC1 and PC2) with eigenvalues greater than λ ≥ 1.0, as highlighted by [16]. The eigenvalues and percentage of variance explained for each component are shown in Table 1. The two components together represented 76.30% of the total variation. PC1 explained 49.59% of the total variance, which is represented by most variables, and PC2 explained 26.71% of the remaining variance.

There was a significant effect (*p* ≤ 0.01) of the interaction between the electrical conductivity of water (ECw) and sour passion fruit genotypes (GEN) for PC1 and PC2 (Table 1). When analyzed separately, significant effects (*p* ≤ 0.01) were observed for ECw and GEN in the two principal components (PC1 and PC2).

Effects of treatments and the variables are expressed in Figure 1A,B, referring to the first and second principal components (PC1 and PC2). In the first principal component (PC1), there is a possible interaction between the levels of electrical conductivity of irrigation water and the sour passion fruit genotypes (ECw × GEN). Correlation coefficients were higher than 0.70 for relative water content, chlorophyll *b*, transpiration, stomatal conductance, CO_2_ assimilation rate, water use efficiency, intrinsic carboxylation efficiency, electron transport rate, and photosynthetically active radiation.

The highest relative water content (90.28%) was observed in the genotype G1 under irrigation water of 2.7 dS m^−1^ (EC4G1), which was equivalent to a 14.89% increase in RWC compared to the lowest value (76.84%) obtained in plants of the same genotype subjected to ECw of 1.1 dS m^−1^ (EC2G1) (Table 1). 

The highest contents of chlorophyll *b* (295.34 μg mL^−1^) were obtained in the genotype G1 subjected to ECw of 1.9 dS m^−1^ (EC3G1), which was 827.28% higher (264.1 μg mL^−1^) than that obtained in plants of the same genotype grown under ECw of 2.7 dS m^−1^ (31.13 μg mL^−1^) (EC4G1). 

Regarding transpiration (E), the genotype G1 irrigated with water of 1.1 dS m^−1^ (EC2G1) reached the highest value (3.77 mmol H_2_O m^−2^ s^−1^), which was 127.10% (2.11 mmol H_2_O m^−2^ s^−1^) higher than that obtained in plants irrigated with water of 1.9 dS m^−1^ of the genotype G2, which reached the lowest value (1.66 mmol H_2_O m^−2^ s^−1^). 

The genotype G2 irrigated with water of 1.1 dS m^−1^ (EC2G2) showed the highest stomatal conductance (0.52 mol H_2_O m^−2^ s^−1^). When compared to the lowest value observed (0.10 mol H_2_O m^−2^ s^−1^) referring to the same genotype (G2) irrigated with water of 1.9 dS m^−1^ (EC3G2), a reduction of 0.42 mol H_2_O m^−2^ s^−1^ was observed. 

The highest CO_2_ assimilation rate (38.38 μmol CO_2_ m^−2^ s^−1^) was observed in the genotype G3 irrigated with ECw of 0.3 dS m^−1^ (EC1G3), which was 76.25% (29.26 μmol CO_2_ m^−2^ s^−1^) higher than that obtained under water salinity of 1.9 dS m^−1^ (EC3G3), equal to 9.11 μmol CO_2_ m^−2^ s^−1^.

Water use efficiency (WUEi) showed a behavior similar to that obtained for CO_2_ assimilation rate (Table 1), with the highest value obtained with the genotype G3 irrigated with water of 0.3 dS m^−1^ (EC1G3) and equal to 11.44 [(μmol CO_2_ m^−2^ s^−1^) (mol H_2_O m^−2^ s^−1^) ^−1^], which was 205.88% (7.69 [(μmol m^−2^ s^−1^) (mol H_2_O m^−2^ s^−1^)^−1^]) higher than that obtained by the same genotype irrigated with water of 1.9 dS m^−1^ (EC3G3). 

For intrinsic carboxylation efficiency (CEi), the genotype G3 irrigated with water of 0.3 dS m^−1^ (EC1G3) obtained the highest value (0.31 [(μmol CO_2_ m^−2^ s^−1^) (μmol CO_2_ m^−2^ s^−1^) ^−1^], which was an increase of 933.33% (0.28 [(μmol CO_2_ m^−2^ s^−1^) (μmol CO_2_ m^−2^ s^−1^) ^−1^] compared to the lowest value observed (0.03 [(μmol CO_2_ m^−2^ s^−1^) (μmol CO_2_ m^−2^ s^−1^)^−1^], referring to the genotype G2 irrigated with water of 1.9 dS m^−1^ (EC3G2).

For the electron transport rate (ETR), the genotype G1 irrigated with water of 1.9 dS m^−1^ (EC3G1) obtained the highest value (77.55), which was 282.01% (57.25) higher than that obtained by the genotype G3 (20.30) irrigated with water of 0.3 dS m^−1^ (EC1G3). Similarly, a reduction of 80.71% (385.00 μmol m^−2^ s^−1^) was observed in the PAR of the genotype G1 irrigated with water of 1.9 dS m^−1^ (EC3G1) compared to the genotype G3 cultivated with water of 0.3 dS m^−1^.

Regarding the results obtained in the principal component 2, the eigenvalues were higher than 0.66 for the following variables: chlorophyll *a*, total chlorophyll, carotenoids, and intercellular CO_2_ concentration. The highest content of chlorophyll *a* (Chl *a*) was observed in the genotype G2 (1439.23 μg mL^−1^) irrigated with water of 3.5 dS m^−1^ (EC5G2), which was 940.88% (1300.96 μg mL^−1^) higher than that obtained by the genotype G2 (138.27 μg mL^−1^) irrigated with water of 2.7 dS m^−1^ (EC4G2). 

Total chlorophyll (Chl *t*) contents showed a behavior similar to that observed for Chl *a* contents (Table 1), for which the highest value was obtained by the genotype G2 (1730.19 μg mL^−1^) irrigated with water of 3.5 dS m^−1^ (EC5G2). This value was 914.06% (1559.57 μg mL^−1^) higher than that obtained by the same genotype irrigated with water of 2.7 dS m^−1^ (EC5G2) and equal to 170.62 μg mL^−1^.

The behavior of carotenoid (Car) contents was also similar to that of Chl *a* contents (Table 1), with the highest value obtained by the genotype G2 (365.84 μg mL^−1^) irrigated with water of 3.5 dS m^−1^ (EC5G2), which was 703,16% higher than that obtained by the same genotype irrigated with water of 2.7 dS m^−1^ (45.55 μg mL^−1^).

For intercellular CO_2_ concentration (Table 1), the highest value (345.50 μmol CO_2_ m^−2^ s^−1^) was obtained by the genotype G2 irrigated with water of 1.9 dS m^−1^ (EC3G2). On the other hand, the lowest value (140.00 μmol CO_2_ m^−2^ s^−1^) was observed in the genotype G3, under irrigation with water of 0.3 dS m^−1^. 

Changes in the behavior of physiological variables for each genotype are presented in Pearson’s correlation matrix (Figure 2), with all genotypes showing a strong positive correlation between the CO_2_ assimilation rate and stomatal conductance (>0.80), as well as with the intrinsic efficiencies of water use (>0.85) and carboxylation (>0.80). Behavior was similar to that obtained for photosynthetic pigments, which maintained a correlation greater than 0.90 between themselves, with a weak correlation to the CO_2_ assimilation rate.

In G1 (Figure 2A), differing from the other genotypes, transpiration presented a strong negative correlation with photosynthetic pigments (<−0.90), similar to that observed for stomatal conductance, which presented a correlation in the range of −0.70, and differing from photosynthetically active radiation, which maintained a positive correlation greater than 0.88 with these variables.

The electron transport rate presented different responses between genotypes, showing the least influence in G2 (Figure 2B), with correlations lower than 0.50 in most variables, except for the negative correlations observed in chlorophyll *b* (−0.57) and carotenoids (−0.68). ETR presented a strong positive correlation (>0.70) with PAR in all genotypes.

Relative water content was negatively correlated with gas exchange variables in all genotypes, except for Ci in G3, which showed a positive correlation of 0.68 (Figure 2C). Furthermore, RWC presented a strong positive correlation with ETR and PAR in genotypes G1 and G3, with no effect on G2.

### 2.2. Cellular Damage and Chlorophyll a Fluorescence in Sour Passion Fruit Genotypes Irrigated with Saline Water

There was an interaction between the ECw × GEN factors for all variables analyzed (Table 2). The electrical conductivity of irrigation water (ECw) significantly affected all variables measured, except for initial fluorescence (F_0_). Electrolyte leakage (%EL) was statistically influenced by sour passion fruit genotypes at 154 DAT.

The increase in the salinity levels of irrigation water promoted increments in %EL of 148.21, 6.12, and 36.57% per unit increase in ECw in the genotypes G1, G2, and G3, respectively (Figure 3A). The genotypes G1, G2, and G3 were subjected to ECw of 3.5 dS m^−1^ and showed increments of 220.60%, 22.06%, and 133.17% in %EL, respectively, compared to those subjected to the lowest ECw level (0.3 dS m^−1^).

When the genotype factor was analyzed as a function of each salinity level (Figure 3A), significant differences in %EL were observed at all ECw levels. When cultivated with waters of 0.3, 1.1, and 1.9 dS m^−1^, the genotype G1 had the lowest %EL (22.03, 27.09, and 28.24%), being statistically inferior to G3 and G2. Under irrigation with water of 2.7 and 3.5 dS m^−1^, the lowest values were obtained by the genotype G2, being statistically inferior to G3 at both ECw levels. It is worth pointing out that, when irrigated with water of 0.3 dS m^−1^, plants did not suffer cell damage because the %EL must be higher than 50% for this to be the case—a behavior not observed under these conditions.

For initial fluorescence (F_0_) (Figure 3B), the genotype G3 obtained the lowest estimated value (377.56) under ECw of 1.7 dS m^−1^ and the highest value (445.75) under ECw of 3.5 dS m^−1^. For the genotypes G1 and G2, the F_0_ data were not satisfactorily described with the tested models (R^2^ < 0.6). A significant difference was observed between the genotypes only when they were subjected to irrigation with ECw from 2.7 dS m^−1^. The genotype G1 obtained the highest mean value (439), being statistically superior to G2 (387) (Figure 3B). When the genotypes were subjected to ECw of 3.5 dS m^−1^, the highest value (459) was obtained with G3, which did not differ from G1, and these were statistically superior to G2.

The maximum fluorescence (Fm) of plants of the genotype G1 irrigated with ECw of 3.5 dS m^−1^ reached the highest estimated value (1803.64) (Figure 3C), whereas the lowest estimated value was obtained under ECw of 1.4 dS m^−1^ (1417.96). For the genotype G3, there was a linear increase of 4.70% per unit increase in ECw. When comparing the Fm of the genotype G3 in plants subjected to ECw of 3.5 dS m^−1^ to the value of those that received 0.3 dS m^−1^, an increase of 17.02% (259) was observed.

For Fm, genotypic variations were observed as a function of ECw levels (Figure 3C). Under ECw of 0.3 dS m^−1^, the genotype G1 was statistically superior to the others, with increments of 26% and 17.72% compared to G2 and G3, respectively. Under ECw of 1.1 and 1.9 dS m^−1^, the genotype G2 stood out in comparison to the others, and its values were 26.24% and 15.5% higher than those of G1 and G3, respectively, when irrigated with water of 1.1 dS m^−1^, and 17.26% higher than that of G1 when irrigated with water of 1.9 dS m^−1^. When the plants were subjected to ECw of 2.7 dS m^−1^, the genotype G3 obtained the highest mean value (1813), which was 23.01% higher than that of G2. There was no statistical differences among the genotypes when irrigated with water of 3.5 dS m^−1^.

For variable fluorescence (Fv), the results obtained with the genotypes G1 and G3 were described using the quadratic regression model (Figure 4A), with the maximum and minimum estimated values of 1384.92 and 1058.75 obtained in plants subjected to ECw of 3.5 dS m^−1^ and 1.8 dS m^−1^, respectively. Similar behavior can be observed for the genotype G3, as the maximum and minimum values were estimated at ECw of 0.3 dS m^−1^ and 2.0 dS m^−1^, which are equivalent to 1353.94 and 1161.94, respectively.

In the decomposition of the interaction for Fv, a significant difference was observed between the genotypes only when they were subjected to ECw of 1.9 dS m^−1^; G2 obtained the highest value (1223), which was equal to the value obtained with G3 and statistically higher than that obtained with G1 (1030), a value that was 15.78% lower than that of G2 (Figure 4A).

The quantum efficiency of photosystem II (Fv/Fm) of the genotypes G2 and G3 were described using the quadratic regression model (Figure 4B), with estimated maximum values of 0.827 and 0.884, respectively, under irrigation with water of 0.3 dS m^−1^. Decomposition of the interaction showed a significant difference between the genotypes only when they were subjected to ECw of 0.3 dS m^−1^ and when G3 obtained the highest value (0.9), which was equal to the value obtained with G2 and statistically higher than that of G1 (0.75), which was 16.66% lower than that obtained with G3 (Figure 4B). 

There was a significant effect of the interaction between the ECw × GEN factors for all variables evaluated, except for leaf temperature (LTemp) (Table 3). The ECw levels influenced all variables, except for LTemp. For the genotypes, a significant effect was observed only on the leaf temperature of sour passion fruit plants at 154 DAT.

The increase in the electrical conductivity of irrigation water caused a linear reduction in the initial fluorescence before the saturation pulse (Fs) of the sour passion fruit genotype G2, which was equal to 26.55% per unit increase of ECw (Figure 5A). The data referring to the genotypes G1 and G3 were not satisfactorily described with the regression models studied.

When the genotype factor was analyzed as a function of each ECw level, it was observed that, in plants irrigated with water of 0.3 dS m^−1^, the highest mean value of Fs was obtained with G2, which was equal to 149 and statistically higher than that of G1 (120), which was 19.46% lower than that obtained with G2 (Figure 5A). When subjected to ECw of 1.9 dS m^−1^, the genotype G3 obtained the highest value (209), being statistically superior to G1 and G2, with increments of 91.74% and 81.73%, respectively. There was no significant difference in Fs when the genotypes were subjected to the other ECw levels.

A reduction in Fms was observed in the sour passion fruit genotypes with an increase in ECw (Figure 5B). The genotypes G1 and G3 were described using the quadratic regression model, with maximum values estimated at 308.48 and 328.87, respectively, under irrigation water salinity of 1.3 dS m^−1^. The genotype G2 showed a decreasing linear behavior with an estimated reduction of 30.84% in Fms when comparing the lowest ECw (0.3 dS m^−1^) to the highest ECw (3.5 dS m^−1^), which corresponded to a decrease of 9.36% per unit increase in ECw.

When the genotype factor was analyzed, considering each ECw level, it was observed that the highest value of Fms (394) was obtained for the genotype G2 for plants irrigated with water of 0.3 dS m^−1^, being statistically superior to G1 and G3, with increments of 55.73% and 34.01%, respectively (Figure 5B). When subjected to ECw of 2.7 dS m^−1^, G2 also obtained the highest value (301.33), but it was statistically superior only to G1. There was no significant difference in Fs when the genotypes were subjected to Ecw of 1.1, 1.9, and 3.5 dS m^−1^.

For the potential quantum efficiency of photosystem II (Y), the genotypes G2 and G3 were described using the quadratic regression model, with maximum values estimated at 0.639 and 0.527 under irrigation of 0.3 dS m^−1^ and 3.5 dS m^−1^, respectively (Figure 5C). The data of the genotype G1 were described using the decreasing linear regression model, with an estimated reduction of 26.32% in Y when comparing the lowest ECw (0.3 dS m^−1^) with the highest ECw (3.5 dS m^−1^), being equivalent to a reduction of 8.02% per unit increase in ECw.

Decomposition of the interaction of genotypes as a function of ECw levels for Y showed that under ECw of 0.3, 1.1, and 2.7 dS m^−1^, the genotypes G1 and G2 were statistically superior to G3, with reductions of 17.74% and 17.72% at ECw of 0.3 dS m^−1^, 31.03%, and 32.20% at ECw of 1.1 dS m^−1^, and 25% and 29.41% at ECw of 2.7 dS m^−1^, respectively (Figure 5C). When subjected to ECw of 1.9 dS m^−1^, the genotype G1 obtained the highest value (0.49), being statistically superior to G3, but not differing from G2. Under ECw of 3.5 dS m^−1^, the highest value (0.57) was obtained with G2, which was statistically superior to G1, but there were no significant differences from the genotype G3.

## 3. Discussion

In plants grown under salt stress conditions, the first physiological response of accumulation of salts in the soil is a reduction of stomatal conductance, this occurs as a way to compensate for the water restriction caused by the osmotic imbalance in the soil, leading to a reduction in leaf transpiration—CO_2_ entry into the substomatal chamber which leads to the inhibition of photosynthesis by the plant [17,18]. This is a behavior that corroborates with the positive correlation between gs, E, and A in this research.

However, as observed in the main components, the inverse relationship between Ci and A may be related to the carbon consumption for the functioning of the Calvin cycle at low salinities. However, this effect is more significant in genotypes 2 and 3, which show a strong correlation between carbon consumption and photosynthesis. Unlike genotype 1, which showed a weak positive correlation between photosynthesis and carbon input, this behavior comes from the high values of gas exchange in the initial salinity, which, although consumes carbon through the activity of Rubisco, keeps the atmospheric carbon input high through the stomata. This process is affected by the accumulation of salts in the soil [8,10].

Similar results were also reported by Lima et al. [19], who evaluated the effects of saline water irrigation management strategies (1.3 and 4.0 dS m^−1^) at different phenological stages of plants and potassium fertilization and found that ECw of 4.0 dS m^−1^ caused reductions in the chlorophyll’s synthesis, transpiration, and instantaneous carboxylation efficiency of ‘BRS GA1’ sour passion fruit plants, regardless of their development stage. A reduction in the photosynthetic efficiency in sour passion fruit plants with increasing salinity levels of irrigation water (0.5 to 4.5 dS m^−1^) was also observed by Souto et al. [20].

In this study, different effects were observed regarding the gas exchange of sour passion fruit genotypes. This is related to the genetic variability present in the genotypes studied, since the intensity of salt stress effects depends on factors such as developmental stage, duration of exposure to stress, irrigation management, edaphoclimatic conditions, and genotypes used, among others [21]. 

Furthermore, in glycophytes, excess salts inhibit protein formation and increase the activity of the chlorophyllase enzyme, which is responsible for the degradation of photosynthetic pigments (besides inducing oxidative stress through the accumulation of reactive oxygen species), negatively affecting photosynthesis [22,23].

This is a behavior that was not observed in the photosynthetic pigments of the passion fruit, with an increase being observed in the contents of chlorophyll *a*, total chlorophyll, and carotenoids and found at the highest electrical conductivities of irrigation water (3.5 dS m^−1^). This fact explains the weak negative correlation of photosynthetic pigments with gas exchange, demonstrating a lower propensity for loss in the plant’s photochemical apparatus concerning the biochemistry of photosynthesis under conditions of saline stress. This may be related to the intrinsic characteristics of the genotype ‘BRS Sol do Cerrado’ and vary according to the tolerance level of the crop. Divergent results regarding the synthesis of chlorophylls in plants subjected to salt stress have been reported by several authors; while some record reductions, others report increments [24]. Paiva et al. [25], while studying the effects of salinity (0.3 and 3.5 dS m^−1^) of irrigation water with distinct cationic natures on ‘BRS Rubi do Cerrado’ sour passion fruit, observed that the use of water with an electrical conductivity of 3.5 dS m^−1^ and sodic nature (Na^+^) promoted the greatest synthesis of chlorophyll *a*, chlorophyll *b*, and carotenoids in the plants at 180 DAT.

Under severe salt stress conditions, the increase in Ci is indicative of the deterioration of the photosynthetic structure, since the damage caused to the structures responsible for CO_2_ fixation is also due to the accumulation of salts in the leaves. The increase in *Ci* observed in plants irrigated with the highest level of electrical conductivity (3.5 dS m^−1^) is an indication that the carbon absorbed by the leaves was not being metabolized by the photosynthetic system due to the low activity of the enzyme ribulose- 1,5-biphostat carboxylase oxygenase (RuBisCO) [26] and due to the effects of salt stress. This due to the osmotic effect, which limits water absorption, and due to which plants tend to close their stomata to reduce water loss to the atmosphere.

This is a behavior that helps explain the increase in %EL, since the change in the soil water potential was caused by salinity results in a condition of physiological drought, nutritional and hormonal imbalances, formation of reactive oxygen species, with these factors leading to the destabilization of the cell membrane, among other effects [27]. The degradation of lipids, caused by the accumulation of reactive oxygen species (ROS), damaged the cell membrane and caused oxidative stress because of an excess of energy being directed to oxygen and thus generating several ROS, including superoxide, hydrogen peroxide, hydroxyl radicals, and singlet oxygen [28].

The decrease in the potential quantum efficiency of photosystem II observed in plants under salt stress indicates that a lower amount of excitation energy was used for the synthesis of ATP and NADPH in chloroplasts [29]. This reduction is often attributed to an imbalance in the electron transport rate and a reduction in ATP and NADPH consumption during the CO_2_ assimilation process in plants [30,31].

In this work, it is noted that there was a considerable variation in photosynthetic variables depending on the treatments. This is possibly related to the variability observed in photosynthetically active radiation (PAR), which may have been caused due to external factors, such as the climatic conditions imposed during the evaluations. This may have occurred since the experiment was carried out in the field and was subject to the action of external factors such as the presence of dust particles under the leaf surface, variations in luminosity caused by the presence of passing clouds, temperature, humidity, and characteristics of edaphoclimatic factors of the region where the study was carried out.

## 4. Materials and Methods

The experiment was conducted from November 2021 to July 2022 in the fruit growing sector, located at the ‘Rolando Enrique Rivas Castellón’ experimental farm, belonging to the Center of Sciences and Agri-Food Technology of the Federal University of Campina Grande, in the municipality of São Domingos, PB, Brazil, whose coordinates are 06°48′50′′ S latitude and 37°56′31′′ W longitude, at an altitude of 190 m.

A randomized block design was used, in a 5 × 3 factorial scheme, whose treatments were obtained by combining 2 factors: 5 levels of electrical conductivity of irrigation water—ECw (0.3, 1.1, 1.9, 2.7, and 3.5 dS m^−1^)—and 3 genotypes of sour passion fruit (Gigante Amarelo—‘BRS GA1’; Sol do Cerrado—‘BRS SC1’; and Catarina—‘SCS 437’) with 3 replicates, totaling 45 experimental units. The ECw levels used were adopted in accordance with a study conducted by Lima et al. [32].

The genotypes ‘BRS Gigante Amarelo’ (‘BRS GA1’) and ‘BRS Sol do Cerrado’ (‘BRS SC1’) were released by the Brazilian Agricultural Research Corporation (EMBRAPA) in 2008. They are characterized by producing fruits with a yellow color and oblong shape, weighing from 120 to 350 g, with a pulp yield of 40% and 38%, and soluble solids contents of 13 to 15° Brix, respectively. ‘BRS SC1’ is tolerant to anthracnose, bacteriosis, and various viruses, and the two genotypes bloom throughout the year, with a greater concentration of them being seen in the driest periods [33,34]. 

The genotype ‘SCS 437’ stands out for its tolerance to bacteriosis and anthracnose, resistance to transport, and excellent visual appearance. Its fruits are large, with a mean weight of 315 g, a juice yield of 33 to 50%, soluble solids content of 9 to 14.5 °Brix, a yellow color, and an orange pulp [35].

The seedlings were formed under protected environmental conditions, arranged on benches located near the experimental area. Polyethylene bags with dimensions of 15 × 20 cm were used as containers and filled with substrate composed of a mixture of soil and aged cattle manure at a ratio of 2:1 (on a volume basis), respectively. 

Three seeds were sown per bag at a 0.5 cm depth. After the emergence of seedlings, thinning was carried out, leaving only one plant per bag. During the formation of the seedlings, irrigation was performed using public-supply water from the municipal network (0.3 dS m^−1^), which is considered the lowest level of electrical conductivity control. 

The experiment was conducted under field conditions using 100 L pots adapted as drainage lysimeters, filled with approximately 110 kg of a Neossolo Flúvico Ta Eutrófico típico (Fluvent) of loamy sand texture from a private property located near the experimental area, whose physical-hydraulic, chemical, and salinity characteristics were determined according to the methodology recommended by Teixeira et al. [36]: Ca^2+^, Mg^2+^, Na^+^, K^+^, Al^3+^ + H^+^ = 3.00; 2.44; 0.05; 0.12; 0.69 cmol_c_ kg^−1^, respectively; pH (soil:water, 1:2.5) = 6.01; ECse = 0.71 dS m^−1^; organic matter = 0.21 dag kg^−1^; sand, silt, and clay = 75.65, 20.21, and 4.34 dag kg^−1^, respectively; *p* = 0.53 mg kg^−1^; SAR = 0.61 (mmol L^−1^)^0.5^; ESP = 0.8%; CEC = 6.25 cmol_c_ kg^−1^.

Plants were trained on a vertical trellis system, built with smooth galvanized steel wire No. 12, fixed at 1.2 m height from the lysimeter soil surface, and guided by nylon strings. When they exceeded 10 cm above the trellis, their apical bud was pruned to stimulate the growth of secondary branches, which were guided in opposite directions until reaching 1.50 m in length. After the secondary branches reached the pre-established length, the apical bud was pruned to stimulate the growth of tertiary branches, which were grown until they were 30 cm away from the soil surface of the experimental area. This was performed in order to avoid possible contamination through the contact with the soil. The spacing adopted was 3.0 m between plants and 2.5 m between rows. 

Seedlings were transplanted to the lysimeters 60 days after sowing, a period characterized by the beginning of tendril growth. During the first 30 days after transplantation (DAT), in order to allow acclimatization under field conditions, all plants were irrigated with water of 0.3 dS m^−1^ and, from 31 DAT, irrigation began to be performed using water with the respective levels of electrical conductivity.

Fertilization with NPK was performed using urea (45% N), single superphosphate (20% P_2_O_5_), and potassium sulfate (51.5% K_2_O) as sources of nitrogen, phosphorus, and potassium, respectively. Fertilization with nitrogen and potassium was split and applied monthly in equal doses throughout the cycle of sour passion fruit, with ratios of 1/1 (N/K) in the flowering stage, 1/2 at the beginning of harvest, and 1/3 until the end of harvest, whereas phosphorus was applied all at once as a basal dose, incorporated into the soil as the lysimeters were filled, as recommended by Costa et al. [37]. 

Application of micronutrients was performed biweekly using Dripsol^®^ micro (Mg^2+^ = 1.1%; B = 0.85%; Cu (Cu-EDTA) = 0.5%; Fe (FeEDTA) = 3.4%; Mn (Mn-EDTA) = 3.2%; Mo = 0.05%; Zn = 4.2%; with 70% of EDTA chelating agent) at a concentration of 1 g L^−1^, with foliar spraying. During the applications of micronutrients, Haiten^®^ adhesive spreader was used at a concentration of 0.15 mL L^−1^.

Irrigation water used in the treatment with the lowest electrical conductivity level (0.3 dS m^−1^) came from a well located in the experimental area of the CCTA/UFCG, and its chemical composition was Ca^+2^, Mg^+2^, K^+^, SO_4_^2−^, HCO_3_^−^, CO_3_^2−^-, Cl^−^ = 0.17; 0.61; 1.41; 0.29; 0.18; 0.81; 0.00; and 1.26 mmol_c_ L^−1^, respectively; electrical conductivity (EC) = 0.30 dS m^−1^; pH = 7.10; sodium adsorption ratio (SAR) = 2.25 (mmol L^−1^)^0.5^.

When necessary, to obtain the higher levels of electrical conductivity, sodium chloride (NaCl) was added to the water from the wells, adjusting the concentrations of the available water considering the relationship between ECw and salt concentration [38]. 

Irrigation was performed using a drip system, with two pressure-compensating drippers with a flow rate of 10 L h^−1^ for each plant. Irrigations were performed daily from 7:00 a.m., applying (in each container) the volume corresponding to that obtained by the water balance, which was determined using Equation (1) as follows:(1)$$VI=\frac{\left(Va-Vd\right)}{\left(1-LF\right)}$$
where VI = volume of water to be applied in the irrigation event (L); Va = volume applied in the previous irrigation event (L); Vd = volume drained (L); and LF = leaching fraction of 0.2, applied every 15 days to reduce the accumulation of salts in the root zone.

Weed control was performed via manual weeding between the rows of drainage lysimeters and around the plant collar. Control of pests and diseases was carried out preventively using chemical products recommended for the crop with doses established by the manufacturers.

Effects of the different treatments were evaluated at 154 DAT, when the plants were in the phenological stage of fruiting, based on photosynthetic pigments, water relations, gas exchange, and chlorophyll *a* fluorescence in the light and dark phase. Contents of photosynthetic pigments were determined according to the methodology of [39].

Gas exchange was evaluated with the CO_2_ assimilation rate—A; transpiration—E; internal CO_2_ concentration—Ci; and stomatal conductance—gs. All of which were determined on the third leaf counted from the apex of the tertiary branch using the LCPro+ portable photosynthesis meter (IRGA) from ADC BioScientific, UK. Ltd (Hoddesdon, England). These data were then used to quantify the intrinsic water use efficiency—WUEi (A/gs)—and instantaneous carboxylation efficiency—CEi (A/Ci). These determinations were performed between 6:30 and 10:00 a.m. on the third fully expanded leaf counted from the apical bud under natural conditions of air temperature, CO_2_ concentration and using an artificial radiation source of 1200 μmol m^−2^ s^−1^. 

Chlorophyll *a* fluorescence was measured based on initial fluorescence (F_0_), maximum fluorescence (Fm), variable fluorescence (Fv), and quantum efficiency of photosystem II (Fv/Fm) in leaves preadapted to the dark using leaf clips for 30 min, between 6:00 and 9:00 a.m., in the median leaf of the intermediate productive branch of the plant, using an OS5p pulse-modulated fluorometer from Opti Science.

Evaluations under conditions using light followed the yield protocol, which were performed using an actinic light source with a multi-flash saturation pulse, coupled to a photosynthetically active radiation determination clip (PAR-Clip), which determined the initial fluorescence before the saturation pulse (Fs), maximum fluorescence after adaptation to the saturating light (Fms), electron transport rate (ETR), potential quantum efficiency of photosystem II (Y), photosynthetically active radiation (PAR), and leaf temperature (LTemp.).

Relative water content (RWC) was determined according to [40] and electrolyte leakage in the leaf blade (%EL) was measured according to [41].

The multivariate structure of the results was evaluated using principal component analysis (PCA), synthesizing the amount of relevant information contained in the original data set in a smaller number of dimensions. This resulted from linear combinations of the original variables generated from the eigenvalues (λ ≥ 1.0) in the correlation matrix, which explained a percentage greater than 10% of the total variance [42].

From the reduction of the dimensions, the original data of the variables of each component were subjected to multivariate analysis of variance (MANOVA) with the Hotelling test [43] at a 0.05 probability level for the levels of electrical conductivity of irrigation water and genotypes of sour passion fruit, as well as for the interaction between them. Only variables with a correlation coefficient greater than or equal to 0.65 were maintained in each principal component (PC) [44]. Statistical analyses were performed using Statistica v. 7.0 software [45].

The data that did not obtain a correlation coefficient above 0.6 were subjected to the Shapiro–Wilk normality test to check for normality. Analysis of variance was applied with the F test (*p* ≤ 0.05) and, when significant, linear and quadratic polynomial regression analysis was performed for the factor of electrical conductivity of irrigation water, while a means comparison test (Tukey’s test, *p* ≤ 0.05) was performed for the sour passion fruit genotypes, using the statistical program SISVAR—ESAL version 5.7 [46]. To verify the relationship between the variables analyzed within the levels of electrical conductivity in the passion fruit genotypes (‘BRS GA1’—Gigante Amarelo—G1; ‘BRS SC1’—Sol do Cerrado—G2; and ‘SCS 437’—Catarina), bivariate analyses were carried out using Pearson’s correlation matrix for each genotype.

## 5. Conclusions

According to the results obtained, it appears that the increase in the electrical conductivity of irrigation water negatively affected most of the physiological characteristics of the sour passion fruit at 154 days after transplanting, causing damage to pigment synthesis, gas exchange, and the photosynthetic efficiency of the studied genotypes. Furthermore, the hypothesis of there being a variation in the degree of tolerance of sour passion fruit genotypes concerning salt stress was confirmed. An increase in the synthesis of photosynthetic pigments (chlorophyll *a*, *b*, *t* and carotenoids) of the ‘BRS Sol do Cerrado’ genotype was found when it was irrigated with water of 3.5 dS m^−1^. Furthermore, there was also an increase in the photosynthetic efficiency of the genotype ‘BRS Gigante Amarelo’ when it was irrigated with water of higher electrical conductivity. Thus, we suggest that new research should be carried out to investigate the effect of the salinity of irrigation water, testing these and other genotypes of sour passion fruit. Trials should preferably be carried out under field conditions with a longer period of exposure of plants to this type of stress.

## Figures and Tables

**Figure 1 plants-12-03894-f001:**
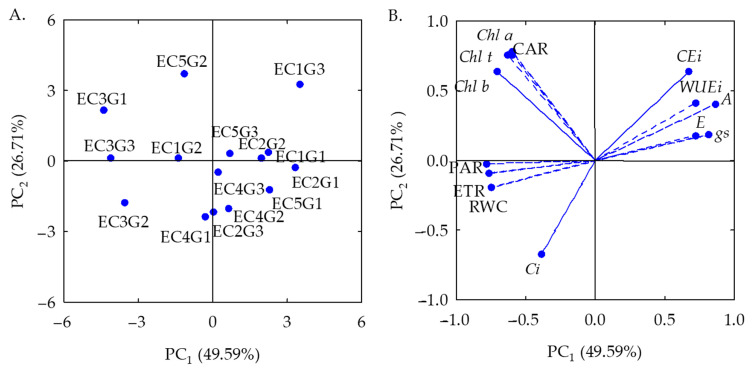
Two-dimensional projection of the scores of the principal components for the factors levels of electrical conductivity of irrigation water (ECw) and genotypes of sour passion fruit (**A**) and of the variables analyzed (**B**) in the two principal components (PC1 and PC2). Electrical conductivity (EC) of water: EC1 (0.3 dS m^−1^); EC2 (1.1 dS m^−1^); EC3 (1.9 dS m^−1^); EC4 (2.7 dS m^−1^); EC5 (3.5 dS m^−1^); G—sour passion fruit genotypes: G1 (‘BRS GA1’—Gigante Amarelo); G2 (‘BRS SC1’—Sol do Cerrado); G3 (‘SCS 437’—Catarina).

**Figure 2 plants-12-03894-f002:**
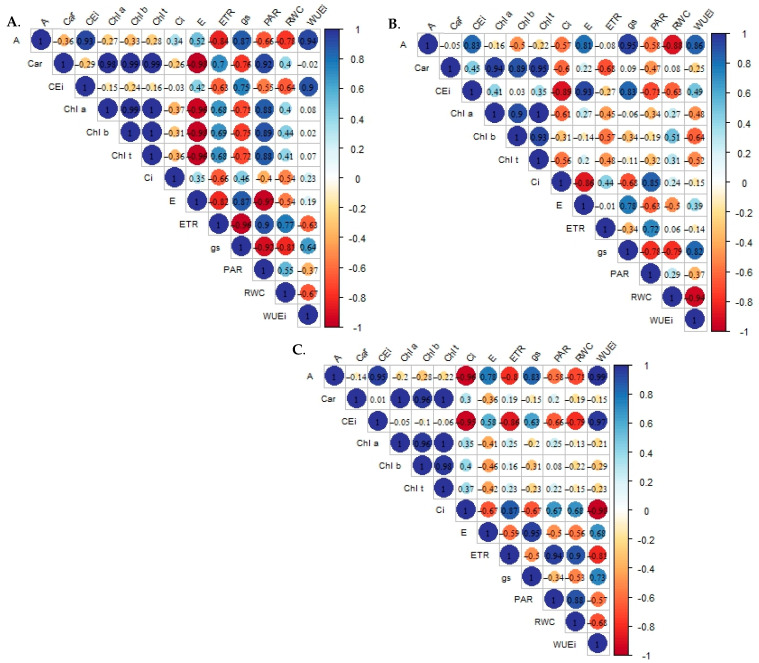
Pearson’s correlation matrix for the variables analyzed within the electrical conductivity levels in the genotypes ‘BRS GA1’—Gigante Amarelo—G1 (**A**); ‘BRS SC1’—Sol do Cerrado—G2 (**B**); and ‘SCS 437’—Catarina—G3 (**C**) at 154 days after transplantation. RWC—relative water content (%); Chl *a*—chlorophyll *a* (μg mL^−1^); Chl *b*—chlorophyll *b* (μg mL^−1^); Chl *t—*total chlorophyll (μg mL^−1^); Car—carotenoids (μg mL^−1^); Ci—intercellular CO_2_ concentration (μmol CO_2_ m^−2^ s^−1^); E—transpiration (mmol H_2_O m^−2^ s^−1^); gs—stomatal conductance (mol H_2_O m^−2^ s^−1^); A—CO_2_ assimilation rate (μmol CO_2_ m^−2^ s^−1^); WUEi—water use efficiency [(μmol CO_2_ m^−2^ s^−1^) (mmol H_2_O m^−2^ s^−1^) ^−1^]; CEi—intrinsic carboxylation efficiency [(μmol CO_2_ m^−2^ s^−1^) (μmol CO_2_ m^−2^ s^−1^)^−1^]; ETR—electron transport rate; PAR—photosynthetically active radiation.

**Figure 3 plants-12-03894-f003:**
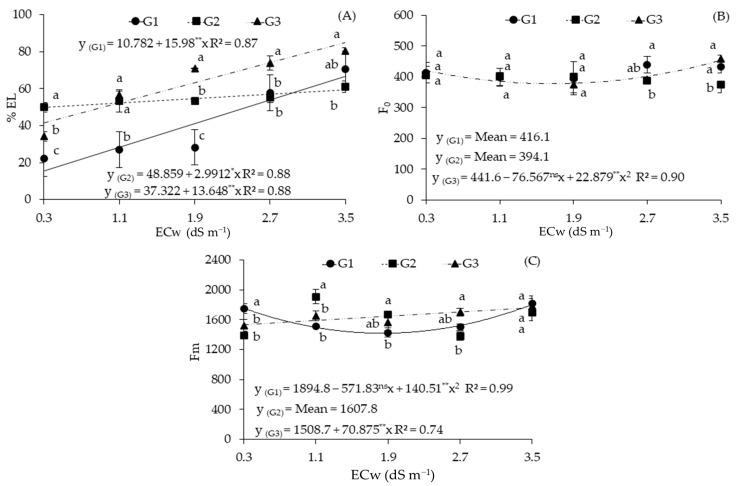
Electrolyte leakage (%EL) (**A**) in the leaf blade; initial fluorescence (F_0_) (**B**); and maximum fluorescence (Fm) (**C**) of sour passion fruit genotypes as a function of the electrical conductivity of irrigation water (ECw) at 154 days after transplantation. Bars with the same letter indicate no significant difference between the means of the sour passion fruit genotypes with Tukey’s test (*p* ≤ 0.05). Sour passion fruit genotypes: G1 (‘BRS GA1’—Yellow Giant); G2 (‘BRS SC1’—Sol do Cerrado); G3 (‘SCS 437’—Catarina). ns, **, and * represent, respectively, non-significant, significant at *p* ≤ 0.01, and *p* ≤ 0.05 with the F test. The vertical lines represent the standard error of the mean (*n* = 3).

**Figure 4 plants-12-03894-f004:**
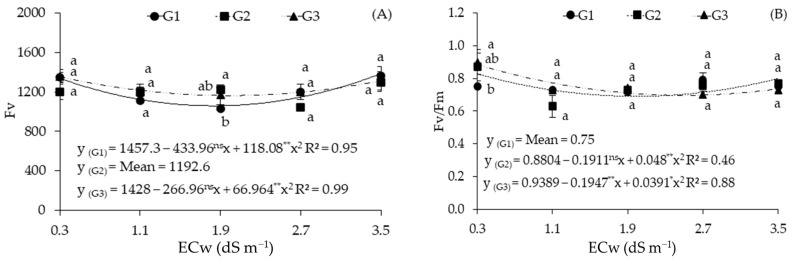
Variable fluorescence (Fv) (**A**) and quantum efficiency of photosystem II (Fv/Fm) (**B**) of sour passion fruit genotypes as a function of the electrical conductivity of irrigation water (ECw) at 154 days after transplantation. Bars with the same letter indicate no significant difference between the means of the sour passion fruit genotypes with Tukey’s test (*p* ≤ 0.05). Sour passion fruit genotypes: G1 (‘BRS GA1’—Yellow Giant); G2 (‘BRS SC1’—Sol do Cerrado); G3 (‘SCS 437’—Catarina). ns, **, and * represent, respectively, non-significant, significant at *p* ≤ 0.01, and *p* ≤ 0.05 with the F test. The vertical lines represent the standard error of the mean (*n* = 3).

**Figure 5 plants-12-03894-f005:**
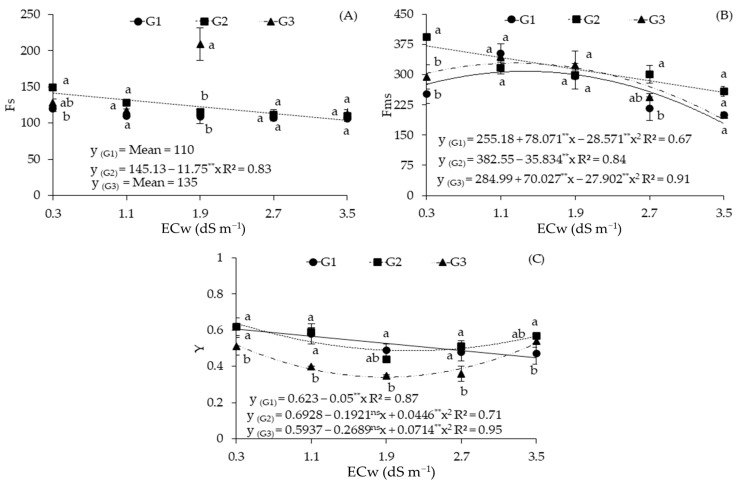
Initial fluorescence before saturation pulse (Fs) (**A**), maximum fluorescence after adaptation to saturating light (Fms) (**B**), and potential quantum efficiency of photosystem II (Y) (**C**) of sour passion fruit genotypes as a function of the electrical conductivity of irrigation water (ECw) at 154 days after transplantation. Bars with the same letter indicate no significant difference between the means of the sour passion fruit genotypes with Tukey’s test (*p* ≤ 0.05). Sour passion fruit genotypes: G1 (‘BRS GA1’—Yellow Giant); G2 (‘BRS SC1’—Sol do Cerrado); G3 (‘SCS 437’—Catarina). ns and ** represent, respectively, non-significant, significant at *p* ≤ 0.01 with the F test. The vertical lines represent the standard error of the mean (*n* = 3).

**Table 1 plants-12-03894-t001:** Eigenvalues, percentage of total variance explained in the multivariate analysis of variance (MANOVA), probability of significance using the Hotelling test (*p* ≤ 0.05) for the factors of the electrical conductivity of irrigation water (ECw), sour passion fruit genotypes (GEN), ECw × GEN interaction, and correlation coefficients (r) between original variables and principal components.

								Principal Components (PCs)					
								PC1			PC2		
Eigenvalues (λ)								6.44			3.47		
Percentage of total variance (S^2^%)								49.59			26.71		
Hotelling test (T^2^) for electrical conductivity (ECw)								0.01			0.01		
Hotelling test (T^2^) for genotypes (GEN)								0.01			0.01		
Hotelling test (T^2^) for ECw × GEN interaction								0.01			0.01		
PCs	Correlation coefficient												
	RWC	Chl *a*	Chl *b*	Chl *t*	Car	Ci	E	gs	A	WUEi	CEi	ETR	PAR
PC1	−0.75	−0.60	−0.70	−0.62	−0.60	−0.39	0.72	0.81	0.87	0.72	0.67	−0.76	−0.78
PC2	−0.19	0.78	0.63	0.76	0.76	−0.66	0.17	0.18	0.40	0.41	0.63	−0.08	−0.02
TREAT	Mean values												
	RWC	Chl *a*	Chl *b*	Chl *t*	Car	Ci	E	gs	A	WUEi	CEi	ETR	PAR
EC1G1	83.98 ± 0.59	537.66 ± 77.61	106.76 ± 2.03	644.43 ± 79.64	106.01 ± 6.11	232.00 ± 9.24	3.60 ± 0.16	0.40 ± 0.01	31.71 ± 0.56	8.87 ± 0.53	0.14 ± 0.01	25.83 ± 2.93	95.00 ± 8.08
EC1G2	89.49 ± 2.67	672.01 ± 61.11	225.95 ± 19.64	897.97 ± 54.03	221.51 ± 5.79	228.00 ± 2.89	2.43 ± 2.89	0.32 ± 0.05	15.64 ± 2.25	6.40 ± 0.76	0.07 ± 0.01	37.40 ± 2.48	153.33 ± 5.49
EC1G3	79.02 ± 4.12	723.31 ± 18.04	154.94 ± 5.19	878.25 ± 23.96	192.45 ± 14.29	140.00 ± 23.09	3.42 ± 23.09	0.34 ± 0.01	38.38 ± 7.03	11.44 ± 2.54	0.31 ± 0.11	20.30 ± 1.96	92.00 ± 1.15
EC2G1	76.84 ± 3.27	289.36 ± 44.45	31.85 ± 3.95	321.21 ± 42.61	52.4 ± 8.04	209.50 ± 2.60	3.77 ± 0.14	0.45 ± 0.04	30.94 ± 2.25	8.29 ± 0.91	0.15 ± 0.01	32.25 ± 1.01	113.33 ± 6.64
EC2G2	81.68 ± 1.18	299.27 ± 10.94	88.37 ± 1.45	387.64 ± 9.48	185.08 ± 46.99	239.00 ± 5.51	2.94 ± 0.20	0.52 ± 0.03	29.21 ± 2.28	9.98 ± 0.78	0.12 ± 0.01	39.40 ± 2.37	174.00 ± 8.08
EC2G3	85.47 ± 2.83	201.6 ± 1.51	43.9 ± 0.21	245.49 ± 1.29	59.6 ± 1.80	230.00 ± 4.62	2.93 ± 0.39	0.17 ± 0.02	15.79 ± 0.12	5.61 ± 0.86	0.07 ± 0.01	39.35 ± 4.24	155.33 ± 7.80
EC3G1	88.16 ± 3.37	1386.47 ± 137.18	295.34 ± 52.70	1681.81 ± 189.88	337.28 ± 22.75	206.00 ± 3.46	2.81 ± 0.31	0.12 ± 0.04	17.87 ± 0.83	6.48 ± 0.58	0.09 ± 0.01	77.55 ± 6.90	477.00 ± 58.31
EC3G2	88.74 ± 4.14	593.98 ± 22.22	215.94 ± 8.632	809.92 ± 30.85	188.32 ± 24.68	345.50 ± 10.68	1.66 ± 0.04	0.10 ± 0.01	9.51 ± 0.14	5.75 ± 0.12	0.03 ± 0.01	43.90 ± 1.91	321.00 ± 15.01
EC3G3	85.96 ± 2.91	1155.18 ± 103.71	275.06 ± 35.36	1430.23 ± 139.07	280.77 ± 33.62	292.00 ± 11.26	2.38 ± 0.12	0.12 ± 0.01	9.11 ± 0.46	3.74 ± 0.33	0.04 ± 0.01	55.50 ± 3.70	260.33 ± 26.85
EC4G1	90.28 ± 0.61	159.14 ± 12.50	31.13 ± 1.07	190.27 ± 10.79	46.63 ± 3.83	209.00 ± 12.12	3.62 ± 0.25	0.24 ± 0.03	15.08 ± 1.05	4.18 ± 0.27	0.07 ± 0.01	58.43 ± 5.82	191.00 ± 42.15
EC4G2	86.53 ± 2.15	138.27 ± 13.18	32.35 ± 4.12	170.62 ± 17.29	45.55 ± 3.21	258.50 ± 4.33	3.10 ± 0.03	0.34 ± 0.02	22.69 ± 0.14	7.33 ± 0.02	0.09 ± 0.01	48.50 ± 1.56	250.33 ± 11.26
EC4G3	87.7 ± 1.35	487.55 ± 0.36	48.54 ± 0.09	536.09 ± 0.44	126.45 ± 1.93	214.00 ± 2.02	3.21 ± 0.04	0.31 ± 0.03	25.18 ± 0.85	7.86 ± 0.25	0.12 ± 0.01	51.80 ± 3.15	287.67 ± 25.12
EC5G1	77.76 ± 5.33	218.43 ± 29.02	49.47 ± 5.88	267.9 ± 35.49	86.51 ± 6.34	274.50 ± 1.44	3.66 ± 0.24	0.39 ± 0.01	25.81 ± 2.38	7.03 ± 0.18	0.09 ± 0.01	24.40 ± 2.31	146.00 ± 6.35
EC5G2	86.52 ± 2.52	1439.23 ± 76.14	290.96 ± 25.16	1730.19 ± 101.30	365.84 ± 20.52	150.50 ± 7.79	3.48 ± 0.22	0.38 ± 0.02	22.13 ± 0.38	6.40 ± 0.43	0.15 ± 0.01	40.30 ± 0.98	167.00 ± 6.35
EC5G3	82.33 ± 1.91	642.39 ± 47.18	138.9 ± 6.57	781.28 ± 53.75	168.52 ± 12,59	231.50 ± 10.10	3.69 ± 0.34	0.37 ± 0.01	24.26 ± 0.08	6.70 ± 0.65	0.11 ± 0.01	40.20 ± 0.75	166.33 ± 7.22

ECw—electrical conductivity of water: EC1 (0.3 dS m^−1^); EC2 (1.1 dS m^−1^); EC3 (1.9 dS m^−1^); EC4 (2.7 dS m^−1^); EC5 (3.5 dS m^−1^); GEN—sour passion fruit genotypes: G1 (‘BRS GA1’—Gigante Amarelo); G2 (‘BRS SC1’—Sol do Cerrado); G3 (‘SCS 437’—Catarina); RWC—relative water content (%); Chl *a*—chlorophyll *a* (μg mL^−1^); Chl *b*—chlorophyll *b* (μg mL^−1^); Chl *t—*total chlorophyll (μg mL^−1^); Car—carotenoids (μg mL^−1^); Ci (intercellular CO_2_ concentration—μmol CO_2_ m^−2^ s^−1^); E—transpiration (mmol H_2_O m^−2^ s^−1^); gs—stomatal conductance (mol H_2_O m^−2^ s^−1^); A—CO_2_ assimilation rate (μmol CO_2_ m^−2^ s^−1^); WUEi (water use efficiency—[(μmol CO_2_ m^−2^ s^−1^) (mmol H_2_O m^−2^ s^−1^)^−1^]; CEi (intrinsic carboxylation efficiency—[(μmol CO_2_ m^−2^ s^−1^) (μmol CO_2_ m^−2^ s^−1^)^−1^]; ETR (electron transport rate); PAR (photosynthetically active radiation). Cells with stronger green and red shades, respectively, represent the highest and lowest average values observed for variables depending on the treatments.

**Table 2 plants-12-03894-t002:** Summary of the analysis of variances for electrolyte leakage (%EL) in the leaf blade, initial fluorescence (F_0_), maximum fluorescence (Fm), variable fluorescence (Fv), and quantum efficiency of photosystem II (Fv/Fm) of sour passion fruit genotypes grown under irrigation water salinity at 154 days after sowing.

Sources of Variation	DF	Mean Squares				
		%EL	F_0_	Fm	Fv	Fv/Fm
Electrical conductivity (ECw)	4	1716.72 **	1312.57 ^ns^	96,165.81 **	65,375.66 **	0.02 **
Linear Regression	1	6809.31 **	664.22 ^ns^	62,357.34 *	2230.04 ^ns^	0.01 ^ns^
Quadratic Regression	1	3.56 ^ns^	4165.87 *	58,933.53 *	24,9600.50 **	0.04 **
Genotypes (GEN)	2	1860.60 **	1857.05 ^ns^	8633.95 ^ns^	13,440.15 ^ns^	0.00 ^ns^
Interaction (ECw × GEN)	8	371.57 **	1713.15 *	87,023.17 **	17,940.35 *	0.00 ^ns^
Blocks	2	7.24	21,774.15	85,552.15	102,639.62	0.00
Residual	28	25.69	581.50	11,812.66	6728.00	0.00
CV (%)		9.57	5.94	6.72	6.76	4.33

DF—degrees of freedom; CV (%)—coefficient of variation; * significant at 0.05 probability level; ** significant at 0.01 probability level; ^ns^ not significant with Tukey’s test (*p* ≤ 0.05).

**Table 3 plants-12-03894-t003:** Summary of the analysis of variance for initial fluorescence before saturation pulse (Fs), maximum fluorescence after adaptation to saturating light (Fms), potential quantum efficiency of photosystem II (Y), and leaf temperature (LTemp) of sour passion fruit genotypes grown under irrigation water salinity at 224 days after sowing.

Sources of Variation	DF	Mean Squares			
		Fs	Fms	Y	LTemp
Electrical Conductivity (ECw)	4	2357.88 **	21,234.08 **	0.03 **	1.31 ^ns^
Linear Regression	1	3074.17 **	67,240.00 **	0.03 **	0.18 ^ns^
Quadratic Regression	1	528.28 ^ns^	12,520.12 **	0.10 **	0.58 ^ns^
Genotypes (GEN)	2	1912.08 **	9881.31 **	0.05 **	0.20 ^ns^
Interaction (ECw × GEN)	8	2146.42 **	4205.16 **	0.00 *	0.08 ^ns^
Blocks	2	401.42	6175.31	0.02	5.22
Residual	28	150.18	914.99	0.00	0.16
CV (%)		10.03	10.55	10.05	1.30

DF—degrees of freedom; CV (%)—coefficient of variation; * significant at 0.05 probability level; ** significant at 0.01 probability level; ns—not significant.

## Data Availability

Data are contained within the article.

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
