# Peer review of "The Effects of Irrigation Water Salinity on the Synthesis of Photosynthetic Pigments, Gas Exchange, and Photochemical Efficiency of Sour Passion Fruit Genotypes"

_plants, 2023, doi:10.3390/plants12223894_

Round 1

Reviewer 1 Report

Comments and Suggestions for Authors

The study is about an interesting and timely subject - the effect of salinity on the photosynthesis of Sour Passionfruit. However, I feel there is work to be done on the interpretation of the results, and possibly more experimental data needs to be collected to reach firm conclusions.

Below are some specific areas (both minor and major) which need work.

Abstract

Line 21 add ‘the’ before present.

Line 28: add ‘An’ before increase

Line 50: are salt levels in this study biologically relevant?

Line 63-64: Describe a little about what these studies showed - were these crops all negatively affected by salt stress?

Results and discussion 

Line 70: The results section needs to start with a sentence or two summarising the experiment, otherwise it can’t be understood without first carefully reading the materials and methods.

Table 1 – In the materials and methods it says 3 replicates were used for each treatment – this table shows the mean values, but standard error is also needed. This is essential to understanding the data.

In order to visualise these data more easily I exported the values and displayed them as a heat map which allows comparison between treatments more easily than just looking at numbers in a table. I would suggest using a similar approach in this table.

The values are very variable – there are no clear patterns in the data – it would be expected that at least some of the variables would increase or decrease with increasing electrical conductivity of the water, but this is not the case. The EC3 treatment for all 3 genotypes does appear to stand out as quite different from the other 4 treatments, which doesn’t make a lot of sense.

It is particularly worrying that PAR values are very varied across the treatments, when this is an external value, and as such should be as close to uniform as possible in all treatments. The particularly high values of PAR in the EC3 treatments could be responsible for those samples having notably different values in the other parameters measured, rather than the differences being due to the changes in electrical conductivity of the water. This either needs to be investigated properly or the work needs repeating controlling the potential confounding variables better.

Line 92 - should read “1A and 1B”

Line 99-101: while the highest RWC is indeed in sample EC4G1, the next highest is in EC1G2. The lowest value is in EC2G1 - with an ECw in between those two treatments, then then next lowest is EC5G2 - so really the RWC values are all over the place, so drawing attention to just the highest and lowest values as though they represent a trend is confusing, and perhaps misleading.

Line 103-105: again, it is confusing to draw attention to 2 datapoints in one genotype, when the relationship between them (highest in EC3 and much lower in EC5) is not true for all genotypes (i.e. G2, where the value for Chl b is actually higher at EC5 than at EC3.

Throughout this section, please refer to genotypes either by their genotype name or (preferably) by the G1-3 code they have been given for easier comparison between tables/figures/text.

Lines 110-114: “A reduction in transpiration with increasing ECw” is not reflected in the numbers, when the transpiration (E) of the EC4 and EC5 treatments are all higher than those at EC3.

These same issues are true throughout this section for the different variables discussed - mostly EC3 stands out as different.

Line 165-166: I feel like this is a big claim considering the variability of the data

The remaining results are based on the same plants  - therefore it is difficult to be sure of the reliability of the dataset.

Line 220: There needs to be a subheading here or a new introductory sentence or two explaining that these next results are about the fluorescence values, and electrolyte leakage, and something explaining what those values tell us. This is really important for aiding understanding of the next section of results.

Table 2: statistical test used needs to be explained in the figure legend

Figure 2: There is a lack of error bars, and statistical tests should be briefly described in the figure legends, with p values given. The letters represent comparisons between genotypes, but that is not immediately clear. Which types of lines have been fitted should be mentioned, and it should be explained why some datasets do not have lines fitted.

Adding the line styles to the legends would be useful.

Statistical tests should also be described within the results section, not just the methods.

Line 253-254 - make it clear the comparison is between the genotypes.

Why is there no mention here of the fact that %EL in the genotype BRS SC1 does not appear to be affected by ECw at all - this is a striking and really interesting result. It could suggest that this genotype has more tolerance of salinity. Where is the text discussing how the individual lines’ %EL are affected by ECw (i.e. the slope of the lines in Figure 2A)? 

Line 259-283: Make it really clear whether the differences being discussed are between genotypes or between ECw treatments - this section is confusing.

Line 284: Again, introduce these variables at some point - why are they interesting?

Table 3: Same comments as Table 2

Figure 3: Same comments as Figure 2

These data seem to reflect the strange findings from Table 1, where the ECw 1.9 stands out from the other treatments - as such I am not sure what conclusions we can draw.

Figure 4: Same comments as Figure 2

Line 336-340: This very high result for SCS 437 at ECw 1.9 appears to be an error - is it pulled in this direction by one individual (error bars would help us to understand this here)? This should be discussed as an anomalous result rather than as an interesting data point.

Materials and Methods

Figure 5 is confusing – relative humidity (%) and precipitation (mm) shouldn’t be on the same axis

Line 391 – again were these values biologically relevant? 

Were the growth conditions and fertilisation regime etc. used relevant to how they are commercially grown? 

Line 485-494: How long was the IRGA allowed to stabilise before measurements were taken?

Conclusions

Line 553-554 - as discussed, this is not reflected in the data

This section feels very brief and it would benefit from referring back to the actual evidence which has helped the authors reach each conclusion.

Comments on the Quality of English Language

English language was good throughout. Slight errors in the abstract.

Author Response

Campina Grande, PB

Nov, 07, 2023

Reference: Plants - 2633396 - Response to Review Report 1

Dear Editor

The authors are grateful to you and the unanimous Reviewers for the positive and constructive comments and suggestions on our manuscript entitled “Effects of Irrigation Water Salinity on the Synthesis of Photosynthetic Pigments, Gas Exchange and Photochemical Efficiency of Sour Passion Fruit Genotypes”. The authors would like to inform you that a thorough revision of the manuscript was made, incorporating the suggestions and adopting the text according to the comments. Attached is the revised version of the manuscript. All changes in the text are highlighted in red color.

The authors remain at your disposal for any further information and explanation.

The responses/clarifications to the issues raised by the Reviewer 1/Editor are presented below:

REVIEWER 1

Comments and Suggestions for Authors

The study is about an interesting and timely subject - the effect of salinity on the photosynthesis of Sour Passionfruit. However, I feel there is work to be done on the interpretation of the results, and possibly more experimental data needs to be collected to reach firm conclusions.

Below are some specific areas (both minor and major) which need work.

Abstract

  1. Line 21 add ‘the’ before present.

Response: As suggested, the term was changed in the revised version of the manuscript.

  1. Line 28: add ‘An’ before increase.

Response: As suggested, the term was changed in the revised version of the manuscript.

  1. Line 50: are salt levels in this study biologically relevant?

Response: Dear Reviewer, passion fruit is a crop that is moderately sensitive to saline stress, mainly depending on the growing conditions, especially those in which the research was carried out (semi-arid conditions). Studies indicate that electrical conductivity levels in irrigation water above 1.3 dS m-1 are already harmful to crops. The saline levels used in this study were based on research carried out by Lima et al. (2020), where they found effects of the salinity of irrigation water (0.3 to 3.0 dS m-1) on several physiological parameters of sour passion fruit cultivation.

Lima, G. S. de; Souza, W. B. B.; Soares, L. A. dos A.; Pinheiro, F. W. A.; Gheyi, H. R.; Oliveira, V. K. N. Dano celular e pigmentos fotossintéticos do maracujazeiro-azedo em função da natureza catiônica da água. Irriga, v.25, n.4, p.663-669, 2020. http://orcid.org/0000-0001-9960-1858

  1. Line 63-64: Describe a little about what these studies showed - were these crops all negatively affected by salt stress?

Response:  Dear Reviewer, were inserted in the revised version of the manuscript genotypic variations the tolerance of crops to the salinity of irrigation water observed in the studies, as can be seen in lines 70-71.

Results and discussion

  1. Line 70: The results section needs to start with a sentence or two summarising the experiment, otherwise it can’t be understood without first carefully reading the materials and methods.

Response: Dear Reviewer, in the results, emphasis was placed on the sentence summarising the experiment, as can be observed in the lines: 83-84 and 205-206

  1. Table 1 – In the materials and methods it says 3 replicates were used for each treatment – this table shows the mean values, but standard error is also needed. This is essential to understanding the data.

Response: As suggested, the standard error (Table 1) was inserted in the revised version of the manuscript.

  1. In order to visualise these data more easily I exported the values and displayed them as a heat map which allows comparison between treatments more easily than just looking at numbers in a table. I would suggest using a similar approach in this table.

Response: As suggested, the heat map (Table 1) was inserted in the revised version of the manuscript.

  1. The values are very variable – there are no clear patterns in the data – it would be expected that at least some of the variables would increase or decrease with increasing electrical conductivity of the water, but this is not the case. The EC3 treatment for all 3 genotypes does appear to stand out as quite different from the other 4 treatments, which doesn’t make a lot of sense.

Response: Dear Reviewer, each variable is presented individually, and there is not necessarily a distinction between the effect of the electrical conductivity of the irrigation water, depending on the genetic variability of the genotypes studied.

  1. It is particularly worrying that PAR values are very varied across the treatments, when this is an external value, and as such should be as close to uniform as possible in all treatments. The particularly high values of PAR in the EC3 treatments could be responsible for those samples having notably different values in the other parameters measured, rather than the differences being due to the changes in electrical conductivity of the water. This either needs to be investigated properly or the work needs repeating controlling the potential confounding variables better.

Response: Dear Reviewer, the variation in PAR values observed in this work is possibly due to external factors that may have contributed to this behavior, such as the presence of dust particles under the leaf surface. Knowing this, in our conclusions we suggested carrying out other research, preferably with a longer experimental period, which could guarantee the carrying out of more evaluations and thus greater reliability of the data collected.

  1. Line 92 - should read “1A and 1B”

Response: As suggested, the term was changed in the revised version of the manuscript.

  1. Line 99-101: while the highest RWC is indeed in sample EC4G1, the next highest is in EC1G2. The lowest value is in EC2G1 - with an ECw in between those two treatments, then then next lowest is EC5G2 - so really the RWC values are all over the place, so drawing attention to just the highest and lowest values as though they represent a trend is confusing, and perhaps misleading.

Response: Dear Reviewer, this variable was evaluated through analysis of main components and multivariate variance and, when we present the data that were subjected to this type of evaluation, the presentation of the results can be carried out highlighting the highest and lowest values observed. Although the values are close, according to the statistical test applied (Hottdling) it was found that there was a significant effect between the treatments, in this sense, we highlight in the results the behavior observed in the variable.

  1. Line 103-105: again, it is confusing to draw attention to 2 datapoints in one genotype, when the relationship between them (highest in EC3 and much lower in EC5) is not true for all genotypes (i.e. G2, where the value for Chl b is actually higher at EC5 than at EC3.

Response: Dear Reviewer, each variable is presented individually, and there is not necessarily a distinction between the effect of the electrical conductivity of the irrigation water, depending on the genetic variability of the genotypes studied.

  1. Throughout this section, please refer to genotypes either by their genotype name or (preferably) by the G1-3 code they have been given for easier comparison between tables/figures/text.

Response: As suggested, BRS GA1 - G1, BRS SC1 - G2 and SCS 437 - G3.

  1. Lines 110-114: “A reduction in transpiration with increasing ECw” is not reflected in the numbers, when the transpiration (E) of the EC4 and EC5 treatments are all higher than those at EC3.

Response: Dear Reviewer, the research was conducted in semi-arid conditions, in this sense, the genotypes behave differently in terms of saline stress conditions, as presented in the work hypothesis.

  1. These same issues are true throughout this section for the different variables discussed - mostly EC3 stands out as different.

Response: Dear Reviewer, the research was conducted in semi-arid conditions, in this sense, the genotypes behave differently in terms of saline stress conditions, as presented in the work hypothesis.

  1. Line 165-166: I feel like this is a big claim considering the variability of the data

Response: Dear Reviewer, the results have been corrected, as can be observed in the line 144.

  1. The remaining results are based on the same plants - therefore it is difficult to be sure of the reliability of the dataset.

Response: Dear Reviewer, the results have been corrected, as can be observed in the lines 370-374.

  1. Line 220: There needs to be a subheading here or a new introductory sentence or two explaining that these next results are about the fluorescence values, and electrolyte leakage, and something explaining what those values tell us. This is really important for aiding understanding of the next section of results.

Response: As suggested, the subheading was added in the revised version of the manuscript.

  1. Table 2: statistical test used needs to be explained in the figure legend

As suggested, statistical test was inserted in the table’s legend.

  1. Figure 2: There is a lack of error bars, and statistical tests should be briefly described in the figure legends, with p values given. The letters represent comparisons between genotypes, but that is not immediately clear. Which types of lines have been fitted should be mentioned, and it should be explained why some datasets do not have lines fitted.

Response: Dear Reviewer, throughout the text, it is explained that the data for some variables did not fit any of the regression models applied, hence the absence of the trend line, and only the description of the treatment accompanied by the average value of the repetitions.

  1. Adding the line styles to the legends would be useful.

Response: As suggested, the line styles to the legends was added in the revised version of the manuscript.

  1. Statistical tests should also be described within the results section, not just the methods.

Response: As suggested, the statistical tests in results section was added in the revised version of the manuscript.

  1. Line 253-254 - make it clear the comparison is between the genotypes.

Response: Dear Reviewer, initially, a comparison is made of each genotype depending on the salinity of the irrigation water, which is the regression analysis, and in a second moment the breakdown of all genotypes at each level of electrical conductivity of the irrigation water is presented. As I have already added the description of the statistical test in each figure, I believe this information is clear.

  1. Why is there no mention here of the fact that %EL in the genotype BRS SC1 does not appear to be affected by ECw at all - this is a striking and really interesting result. It could suggest that this genotype has more tolerance of salinity. Where is the text discussing how the individual lines’ %EL are affected by ECw (i.e. the slope of the lines in Figure 2A)?

Response: Dear Reviewer, it is considered that there has been damage to the cell membrane when the %EL exceeds 50%, a fact observed in all genotypes with the increase in irrigation electric conductivity. The BRS SC1 genotype, despite having presented the lowest %EL value at the level of 3.5 dS m-1, reached 60.86% damage, that is, it was also affected by salinity, despite being statistically inferior only to genotype 3 (SCS 437).

  1. Line 259-283: Make it really clear whether the differences being discussed are between genotypes or between ECw treatments - this section is confusing.

Response: As suggested, the results of all variables, first the breakdown of each genotype within the 5 saline levels is presented, then the breakdown of the genotypes within each level of electrical conductivity is presented.

  1. Line 284: Again, introduce these variables at some point - why are they interesting?

Response: As suggested, the variables in results section was added in the revised version of the manuscript.

  1. Table 3: Same comments as Table 2

Response: As suggested, statistical test was inserted in the table’s legend.

  1. Figure 3: Same comments as Figure 2

Response: As suggested, the line styles to the legends was added in the revised version of the manuscript.

  1. These data seem to reflect the strange findings from Table 1, where the ECw 1.9 stands out from the other treatments - as such I am not sure what conclusions we can draw.

Response: Dear Reviewer, the research was conducted in semi-arid conditions, in this sense, the genotypes behave differently in terms of saline stress conditions, as presented in the work hypothesis.

  1. Figure 4: Same comments as Figure 2

Response: As suggested, the line styles to the legends was added in the revised version of the manuscript.

  1. Line 336-340: This very high result for SCS 437 at ECw 1.9 appears to be an error - is it pulled in this direction by one individual (error bars would help us to understand this here)? This should be discussed as an anomalous result rather than as an interesting data point.

Response: As suggested, the standard error was inserted in the revised version of the manuscript.

Materials and Methods

  1. Figure 5 is confusing – relative humidity (%) and precipitation (mm) shouldn’t be on the same axis

Response: Figure removed

  1. Line 391 – again were these values biologically relevant?

Response: Dear Reviewer, passion fruit is a crop that is moderately sensitive to saline stress, mainly depending on the growing conditions, especially those in which the research was carried out (semi-arid conditions). Studies indicate that electrical conductivity levels in irrigation water above 1.3 dS m-1 are already harmful to crops. The saline levels used in this study were based on research carried out by Lima et al. (2020), where they found effects of the salinity of irrigation water (0.3 to 3.0 dS m-1) on several physiological parameters of sour passion fruit cultivation.

Lima, G. S. de; Souza, W. B. B.; Soares, L. A. dos A.; Pinheiro, F. W. A.; Gheyi, H. R.; Oliveira, V. K. N. Dano celular e pigmentos fotossintéticos do maracujazeiro-azedo em função da natureza catiônica da água. Irriga, v.25, n.4, p.663-669, 2020. http://orcid.org/0000-0001-9960-1858

  1. Were the growth conditions and fertilization regime etc. used relevant to how they are commercially grown?

Response: Dear Reviewer, the management practices used throughout the development of the crop were carried out in accordance with the appropriate technical recommendations, including pruning, fertilization, staking, pest and disease control, in accordance with what was proposed by Costa et al. (2008).

Costa, A. de F.S.; Costa, A.N.; Ventura, J.A.; Fanton, C.J.; Lima, I. de M.; Caetano, L.C.S.; Santana, E.N. de. Recomendações técnicas para o cultivo do maracujazeiro. Vitória, ES: Incaper (Incaper. Documentos, 162). 2008. 56p.

  1. Line 485-494: How long was the IRGA allowed to stabilise before measurements were taken?

Response: Dear Reviewer, test readings were performed in a separate file. Once extreme variations in the test readings were no longer observed, a new file was created, and final evaluations began. This time is approximately 15 to 20 minutes.

Conclusions

  1. Line 553-554 - as discussed, this is not reflected in the data. This section feels very brief and it would benefit from referring back to the actual evidence which has helped the authors reach each conclusion.

Response: As suggested, the conclusions was changed in the revised version of the manuscript.

Yours sincerely,

Geovani Soares de Lima

Reviewer 2 Report

Comments and Suggestions for Authors

Review on the manuscript Effects of Irrigation Water Salinity on the Synthesis of Photosynthetic Pigments, Gas Exchange and Photochemical  Efficiency of Sour Passion Fruit Genotypes

 The world's freshwater resources are limited, and using saline water for irrigation is a perspective approach. And in this regard, the study of physiological mechanisms of salt tolerance in plants is a relevant task. This manuscript focuses on changes in photosynthesis (gas exchange, photosynthetic pigments, photosynthetic light reactions) of three genotypes Passiflora edulis in response to saline water irrigation.

 However, the structure and illustrative material manuscript are not successful and, as presented, the article is very difficult to understand.

 I highly recommend dividing the results description and discussion into two separate sections, where the first will contain a clear, logical description of the results obtained, and the second will contain an analysis of the main results with the available literature data.

 Table 1 “Mean values” - there are no errors and significant differences. It is better to present the data as diagrams with mean (preferably for three repetitions together) ±error (bars) and designations of the significance of differences.

 Figure 1 B – it is better to conduct the multiple correlations for each genotype separately - differences between genotypes may be revealed

 In Fig. 2,3,4 bars (errors) should be added

 At the end of the main text there is missing a summary paragraph with the main results for each genotype.

 Conclusions are a list of the main results. Authors should formulate conclusions based on these results and propose perspectives.

The abstract contains unnecessary methodological details, but there are no conclusions.

Author Response

Campina Grande, PB

Nov, 07, 2023

Reference: Plants - 2633396 - Response to Review Report 2

Dear Editor

The authors are grateful to you and the unanimous Reviewers for the positive and constructive comments and suggestions on our manuscript entitled “Effects of Irrigation Water Salinity on the Synthesis of Photosynthetic Pigments, Gas Exchange and Photochemical Efficiency of Sour Passion Fruit Genotypes”. The authors would like to inform you that a thorough revision of the manuscript was made, incorporating the suggestions and adopting the text according to the comments. Attached is the revised version of the manuscript. All changes in the text are highlighted in red color.

The authors remain at your disposal for any further information and explanation.

The responses/clarifications to the issues raised by the Reviewer 2/Editor are presented below:

REVIEWER 2

Comments and Suggestions for Authors

Review on the manuscript Effects of Irrigation Water Salinity on the Synthesis of Photosynthetic Pigments, Gas Exchange and Photochemical  Efficiency of Sour Passion Fruit Genotypes.

The world's freshwater resources are limited, and using saline water for irrigation is a perspective approach. And in this regard, the study of physiological mechanisms of salt tolerance in plants is a relevant task. This manuscript focuses on changes in photosynthesis (gas exchange, photosynthetic pigments, photosynthetic light reactions) of three genotypes Passiflora edulis in response to saline water irrigation.

However, the structure and illustrative material manuscript are not successful and, as presented, the article is very difficult to understand.

  1. I highly recommend dividing the results description and discussion into two separate sections, where the first will contain a clear, logical description of the results obtained, and the second will contain an analysis of the main results with the available literature data.

Response: As suggested, the results description and discussion into two separate sections in the revised version of the manuscript.

  1. Table 1 “Mean values” - there are no errors and significant differences. It is better to present the data as diagrams with mean (preferably for three repetitions together) ±error (bars) and designations of the significance of differences.

Response: As suggested, the standard error (Table 1) was inserted in the revised version of the manuscript.

  1. Figure 1 B – it is better to conduct the multiple correlations for each genotype separately - differences between genotypes may be revealed

Response: As suggested, the multiple correlations (Figure 2) was inserted in the revised version of the manuscript.

  1. In Fig. 2,3,4 bars (errors) should be added

Response: As suggested, the standard error (Fig. 2,3,4) was inserted in the revised version of the manuscript.

  1. At the end of the main text there is missing a summary paragraph with the main results for each genotype.

Response: Dear Reviewer, in the results, was placed the results for each genotype, as can be observed in the lines: 192-204.

  1. Conclusions are a list of the main results. Authors should formulate conclusions based on these results and propose perspectives.

Response: As suggested, the conclusions was changed in the revised version of the manuscript.

  1. The abstract contains unnecessary methodological details, but there are no conclusions.

Response: As suggested, the abstract was changed in the revised version of the manuscript.

Yours sincerely,

Geovani Soares de Lima

Reviewer 3 Report

Comments and Suggestions for Authors

The authors pf the publication present a well done analysis of salt stress on various passion fruit genotypes.

The statistical analysis is well done and presented. Work sounds well done with appropriate controls and repetitions.

Overall the studies can present some interest for agronomical studies.

Author Response

Campina Grande, PB

Nov, 07, 2023

Reference: Plants - 2633396 - Response to Review Report 3

Dear Editor

The authors are grateful to you and the unanimous Reviewers for the positive and constructive comments and suggestions on our manuscript entitled “Effects of Irrigation Water Salinity on the Synthesis of Photosynthetic Pigments, Gas Exchange and Photochemical Efficiency of Sour Passion Fruit Genotypes”. The authors would like to inform you that a thorough revision of the manuscript was made, incorporating the suggestions and adopting the text according to the comments. Attached is the revised version of the manuscript. All changes in the text are highlighted in red color.

The authors remain at your disposal for any further information and explanation.

The responses/clarifications to the issues raised by the Reviewer 3/Editor are presented below:

REVIEWER 3

Comments and Suggestions for Authors

The authors pf the publication present a well done analysis of salt stress on various passion fruit genotypes.

The statistical analysis is well done and presented. Work sounds well done with appropriate controls and repetitions.

Overall the studies can present some interest for agronomical studies.

Response: We would like to thank the reviewer for the message and comments on the manuscript.

Yours sincerely,

Geovani Soares de Lima

Reviewer 4 Report

Comments and Suggestions for Authors

The paper is well-written and I have no significant complaints or recommendations. The materials and methods are succinctly explained, yet thorough. The only criticism pertains to the conclusions, which could benefit from more elaborate recommendations as a research contribution and potential avenues for future research regarding the discussed issue. Furthermore, the conclusion bears a striking resemblance to the abstract, suggesting an opportunity for enhancing its depth and overall quality.

Author Response

Campina Grande, PB

Nov, 07, 2023

Reference: Plants - 2633396 - Response to Review Report 4

Dear Editor

The authors are grateful to you and the unanimous Reviewers for the positive and constructive comments and suggestions on our manuscript entitled “Effects of Irrigation Water Salinity on the Synthesis of Photosynthetic Pigments, Gas Exchange and Photochemical Efficiency of Sour Passion Fruit Genotypes”. The authors would like to inform you that a thorough revision of the manuscript was made, incorporating the suggestions and adopting the text according to the comments. Attached is the revised version of the manuscript. All changes in the text are highlighted in red color.

The authors remain at your disposal for any further information and explanation.

The responses/clarifications to the issues raised by the Reviewer 4/Editor are presented below:

REVIEWER 4

Comments and Suggestions for Authors

The paper is well-written and I have no significant complaints or recommendations. The materials and methods are succinctly explained, yet thorough. The only criticism pertains to the conclusions, which could benefit from more elaborate recommendations as a research contribution and potential avenues for future research regarding the discussed issue. Furthermore, the conclusion bears a striking resemblance to the abstract, suggesting an opportunity for enhancing its depth and overall quality.

Response: As suggested, the conclusions was changed in the revised version of the manuscript.

Yours sincerely,

Geovani Soares de Lima

Round 2

Reviewer 1 Report

Comments and Suggestions for Authors

This revised version of the manuscript is much improved. the addition of Figure 2 is very useful. I still feel that there should be a comment in the text about the variation in PAR (which is understandable given that the work was carried out in the field), which is clearly likely to have affected photosynthetic variables.

Author Response

This revised version of the manuscript is much improved. the addition of Figure 2 is very useful. I still feel that there should be a comment in the text about the variation in PAR (which is understandable given that the work was carried out in the field), which is clearly likely to have affected photosynthetic variables.

Accepted request. Lines 427 to 424.

Reviewer 2 Report

Comments and Suggestions for Authors

The authors significantly revised the article and took into account most of the comments and recommendations.

Minor comments:

In Table 1 the authors should add explanation of red and green colors

Line 396-397 – reference 26 did not contains study of internal CO2 concentration and activity of RuBisCO. The authors should check it and other links.

Author Response

The authors significantly revised the article and took into account most of the comments and recommendations.

Minor comments:

In Table 1 the authors should add explanation of red and green colors.

Accepted request. Lines 107 to 109.

Line 396-397 – reference 26 did not contains study of internal CO2 concentration and activity of RuBisCO. The authors should check it and other links.

Changed in text. In addition to changing the reference, new discussions were also included in this part. Lines 395 to 399. The previously cited reference was removed and the new reference was inserted.